# Prostate Cancer Disparity, Chemoprevention, and Treatment by Specific Medicinal Plants

**DOI:** 10.3390/nu11020336

**Published:** 2019-02-04

**Authors:** Clement G. Yedjou, Ariane T. Mbemi, Felicite Noubissi, Solange S. Tchounwou, Nole Tsabang, Marinelle Payton, Lucio Miele, Paul B. Tchounwou

**Affiliations:** 1Natural Chemotherapeutics Research Laboratory, NIH/NIMHD RCMI-Center for Environmental Health, College of Science, Engineering and Technology, Jackson State University, 1400 Lynch Street, Jackson, MS 39217, USA; arianeng6@gmail.com; 2Department of Biology, College of Science, Engineering and Technology, Jackson State University, 1400 Lynch Street, Jackson, MS 39217, USA; felicite.noubissi_kamdem@jsums.edu; 3Department of Biology, University of Mississippi, 214 Shoemaker Hall, P.O. Box 1848, MS 38677, USA; 0solanget@gmail.com; 4Department of Animal Biology, Higher Institute of Environmental Sciences, Yaounde P.O.Box 16317, Cameroon; tsabang2001@yahoo.fr; 5Center of Excellence in Minority Health and Health Disparities, School of Public Health, Jackson State University, Jackson Medical Mall-Thad Cochran Center, 350 West Woodrow Wilson Avenue, Jackson, MS 39213, USA; marinelle.payton@jsums.edu; 6Department of Genetics, LSU Health Sciences Center, School of Medicine, 533 Bolivar Street, Room 657, New Orleans, LA 70112, USA; lmiele@lsuhsc.edu

**Keywords:** prostate cancer, health disparity, medicinal plant, chemoprevention

## Abstract

Prostate cancer (PC) is one of the most common cancers in men. The global burden of this disease is rising. Its incidence and mortality rates are higher in African American (AA) men compared to white men and other ethnic groups. The treatment decisions for PC are based exclusively on histological architecture, prostate-specific antigen (PSA) levels, and local disease state. Despite advances in screening for and early detection of PC, a large percentage of men continue to be diagnosed with metastatic disease including about 20% of men affected with a high mortality rate within the African American population. As such, this population group may benefit from edible natural products that are safe with a low cost. Hence, the central goal of this article is to highlight PC disparity associated with nutritional factors and highlight chemo-preventive agents from medicinal plants that are more likely to reduce PC. To reach this central goal, we searched the PubMed Central database and the Google Scholar website for relevant papers. Our search results revealed that there are significant improvements in PC statistics among white men and other ethnic groups. However, its mortality rate remains significantly high among AA men. In addition, there are limited studies that have addressed the benefits of medicinal plants as chemo-preventive agents for PC treatment, especially among AA men. This review paper addresses this knowledge gap by discussing PC disparity associated with nutritional factors and highlighting the biomedical significance of three medicinal plants (curcumin, garlic, and *Vernonia amygdalina*) that show a great potential to prevent/treat PC, as well as to reduce its incidence/prevalence and mortality, improve survival rate, and reduce PC-related health disparity.

## 1. Introduction

Prostate cancer (PC) is one of the most common cancers in men and its incidence continues to rise in many developed countries [1]. Each year, PC causes nearly 30,000 deaths and 230,300 new cases in the United States with the highest incidence and mortality rates among African-American (AA) men [2]. The treatment options for PC care include surgery, radiotherapy, and chemotherapy. However, surgery and radiotherapy are effective when the tumor is localized and detected during the early stages. On the other hand, chemotherapy, while effective during the early stage tumor development, often induces side effects. All these treatment strategies are ineffective once the tumor has metastasized from the prostate to other parts of the human body. The majority of PC deaths may be due to metastases that are highly resistant to current treatment options [3,4]. Even when PC is treated, it usually comes back in the first few years post-treatment.

Androgens play a crucial role in the differentiation, development, and normal functioning of prostate and, therefore, androgens may have an important role in developing prostate carcinogenesis. Androgen deprivation therapy (ADT) administered though surgical or medical castration is currently the standard treatment for men with locally advanced or metastatic prostate cancer [5]. Despite an initial benefit of ADT alone or in combination with docetaxel chemotherapy, some prostate cancer patients will progress to an advanced disease state termed as castration-resistant prostate cancer (CRPC) [6]. CRPC is a lethal form of advanced disease for which treatment options are limited. It depends on the androgen receptor (AR) function for which AR targeting remains the main therapeutic intervention [6,7,8]. For most metastatic PC, ADT is usually not effective because prostate tumors may regrow over time. In addition, ADT is often associated with several side effects that often influence quality of life. The common side effects associated with ADT in men with metastatic cancer include hot flashes, metabolic syndrome, osteoporosis, sexual dysfunction, gynecomastia, depression, and memory difficulties [9,10,11,12]. Clinical trials demonstrated that a new generation of anti-androgen or novel inhibitor targeting androgen synthesis is effective for post-docetaxel CRPC, which suggests that CRPC at that stage is still, at least in part, dependent on the androgen/AR axis [13,14]. 

Since the treatment strategies for metastasized or locally advanced stages of PC are limited, many PC patients rely on traditional medicine as viable therapeutic options [15]. Several neoplastic drugs or chemotherapeutic agents used to treat PC patients have various side effects [16]. Vegetables and fruits are the best known anti-cancer agents that contain a wide variety of different micronutrients with properties that could make it more difficult for cancer to develop. These micronutrients are polyphenols, flavonoids, carotenoids, vitamins, minerals, and other phytochemicals [17]. Eating vegetables and fruits containing high levels of polyphenols and flavonoids contributes to PC prevention. According to several studies and a report from our lab, a poor diet may contribute to approximately 10% to 75% of various cancer-related deaths (Figure 1) [18]. As seen in Figure 1, if a man is eating a healthy diet rich in vegetables and fruits, he can reduce his risk of getting PC by 75%. 

A study reported that Asian populations have a relatively low incidence rate of PC compared to whites and black Americans because they use the extract of medicinal plants against cancer [19]. In general, scientific evidence from epidemiological studies suggests that consumption of high fiber, lean protein, and low fat together with high vegetables and fruits significantly reduces the overall cancer risks [20,21]. However, there is a strong scientific report indicating that nutrients in fruits and vegetables may not reduce the risk of cancer when consumed as supplements. These supplements are harmful and have toxic effects if taken at high doses [22,23]. Many people living in developing countries have been using medicinal plants for many years to maintain their health and meet their primary healthcare requirements due to limited access to medical doctors, poor socioeconomic status, limited screening, and a shortage of nurses [24,25,26]. The phytochemicals present in medicinal plants contain anticancer properties that help kill tumor cells, boost the human immune system, and promote good health [27]. Taxol and polyphenols identified from medicinal plant species are currently used in clinical settings to treat cancer. Scientific data have demonstrated that medicinal plants prevent cancer by creating an unfavorable environment for cancer cells to grow, by preventing the recurrence of cancer, by improving the body’s immune system, and by reducing the side effects of conventional treatments [28,29]. There are limited scientific reports that have addressed the benefits of medicinal plants as chemo-preventive agents for PC especially among AA men. This review paper addresses this knowledge gap by discussing PC disparity associated with nutritional factors and highlighting the biomedical significance of three medicinal plants (curcumin, garlic, and *Vernonia amygdalina*) that possess anti-cancer properties capable of preventing PC development and progression. These chemo-preventive agents have the potential to prevent PC development, as well as reduce its incidence and mortality rates, improve the survival rate, and reduce PC-related health disparity.

## 2. Approaches

The study was carried out to review and discuss scientific information on PC disparity in the United States, and to highlight biomedical significance of three medicinal plants that have been consumed for many years for the prevention and/or treatment of cancer and other diseases. We performed a search on the PubMed Central (https://www.ncbi.nlm.nih.gov/pmc/) database and the Google Scholar website. Our searches included a combination of key terms such as: “medicinal plant, prostate cancer prevention, racial disparity, and AA men”. From the key term searches, we were able to identify peer-reviewed articles that have reported invaluable information about edible medicinal plant products and their application towards the prevention and/or treatment of PC. We only reviewed recently published scientific data and selected works cited between 2000 and 2017. 

## 3. Results and Discussions

This review paper highlights, for the first time, the most commonly used edible medicinal plants as natural chemo-preventive agents for preventing and/or treating PC. Developing countries have fewer reported cases of PC while developed countries like the United States have higher reported cases of PC with the largest incidence and mortality among black men worldwide. Humans have relied on natural plant products to prevent and/or treat diseases for 1000 years and we use this idea as the starting point to investigate the medicinal properties of curcumin, garlic, and *Vernonia amygdalina* for preventing and/or treating PC. Based on the scientific data available in the literature, curcumin, garlic, and *Vernonia amygdalina* are promising chemo-preventive agents. They possess a wide range of health benefits including the maintenance of good health, the maintenance of a healthy immune system function, and the prevention and/or treatment of the disease. PC patients living in both poor and wealthy countries frequently used curcumin, garlic, and/or *Vernonia amygdalina* to prevent and treat cancer and other diseases. In the following discussion, we focused on PC disparity associated with nutritional factors and chemo-preventive effects of the three medicinal plants.

## 4. Prostate Cancer Disparity in the United States

Prostate cancer (PC) is the second leading cause of non-cutaneous cancer-related deaths among males living in the United States [30]. Despite an increased emphasis on early detection through prostate specific antigen (PSA) screening, advanced early treatment, and improved understanding of the prostate cancer risk factors, the disparity remains. AA men have continued to have the highest mortality and lowest survival of prostate cancer, and all cancers combined [31,32,33]. On the contrary, the mortality rate of PC has consistently declined among white men and other ethnic groups in the United States [34]. The overall disparity in cancer between AA and Caucasian men accounts to about 40% of PC [35]. The possible reasons for this racial/ethnic disparity are not completely understood. However, factors such as race, genetics, age, poor diet, and physical inactivity may contribute to the occurrence of PC. In the following sessions, we discuss nutritional factors (diet, obesity, tobacco, alcohol, and physical exercise) that are associated with a socioeconomic status for PC, which are more likely to affect AA men when compared to any other ethnic groups. Figure 2 shows nutritional risk factors that are more likely to contribute to PC disparity.

### 4.1. Dietary Factor in Prostate Cancer Disparity

Poor diet and obesity have long been considered as possible risk factors for PC. Several lines of research have shown the association between animal fat such as red meat consumption and diagnosis of PC especially among AA men [36,37]. The Asian population living in Asia and the United States exhibit the lowest frequencies of PC because they commonly consume soybeans [38]. The organic compounds (isoflavones) that are present in soybeans are thought to have a potential protective effect against PC [39]. The purified form of soy isoflavones are tested in a clinical trial as a potential dietary agent against the PC burden among white and African-American men in North America [40]. By testing the hypothesis that differences in PC incidence among different ethnic groups may be due to dietary factors, scientists have investigated the impact of diet and obesity on risk in a population-based, case-control study. In this study, they recruited 3162 men at nine military medical centers in the United States including men who have prostate cancer and have a radical prostatectomy, and pathological data such as the stage of cancer and the Gleason score, age, race, height, weight, and PSA level. They found that more AA men are obese when compared to white men. AA men were also more likely to have a higher PSA level, more advanced cancers, and positive surgical margins compared to white men [41]. In another study, a group of researchers in Michigan looked at a link between prostate cancer and the metabolic syndrome. They recruited 637 prostate cancer patients and 244 control subjects. They found that the metabolic syndrome was only marginally-related to an increase of PC in AA [42]. Several studies have provided scientific evidence showing that obesity is associated with high-grade PC, PC progression, and PC mortality [43,44,45]. A study conducted by the Prostate Cancer Prevention Trial among 10,258 men undergoing biopsy found that obesity increases the risk of high-grade PC but decrease the risk of low-risk grade PC [43]. In addition, obesity has been shown to be correlated with lower rates of PC relapse-free survival and poor tumor characteristics among black but not white men in the United States [44]. Although several studies have shown that obesity is associated with an increased risk of prostate cancer [46,47], other studies did not find an association between obesity and PC development [48,49,50,51]. A cohort study of 69,991 men found that those with high body mass index (BMI) are at increased risk of high grade and fatal PC. However, BMI is less likely to be associated with low grade PC [52]. Giovannucci et al. (2003) showed an inverse relationship between obesity and PC risk, but only for men under 60 years of age who have a family history of cancer [53]. 

### 4.2. Tobacco and Alcohol Factors in Prostate Cancer Disparity

Like many malignancies, tobacco and alcohol are well-known risk factors for PC. Studies indicated that cigarette smoking is associated with high-grade PC, incidence, and mortality among men [54,55]. However, there are limited studies that have focused on AA men. A study from the Centers for Disease Control and Prevention (CDC) indicated that AA and Caucasian men both have a 21% prevalence of cigarette smoking [56], but AA men have lower rates of heavy smoking and lower rates of smoking cessation [57,58,59,60]. A multi-ethnic study by Adams et al. (2013) found that about 21% of AA men were heavy smokers compared to 30.5% of Caucasian men [61]. There are limited studies that have addressed the association between alcohol drinking and PC disparity. However, heavy alcohol drinking is associated with breast cancer, colon cancer, esophageal cancer, larynx cancer, oraparyngeal cancer, liver cancer, and pancreatic cancer [62,63]. 

### 4.3. Physical Activity Factor in Prostate Cancer Disparity

A study by Clarke and Whittemore (2000) indicated that AA men who are physically inactive are 3.6 more likely to be diagnosed with PC compared to AA men who are physically active [64]. However, physical activity has not been shown to lower the incidence of PC among Caucasian men [65,66]. This significant racial disparity may be due to the presence of lower levels of tumor promoting growth factors among AA men who exercise.

### 4.4. Geographic Factor in Prostate Cancer Disparity 

Diet is considered to play a crucial role in the geographical differences in PC incidence and mortality. The incidence rate of PC in Africa where most people consume high plant-based food is approximately four times lower when compared to the United States where most people consume high protein and fatty dairy products [2,66]. For example, the number of new cases of PC was 52 per 100,000 men in 2012 in Africa while in the United States; the number of new cases of PC was 233 per 100,000 during the same period. The reason for the highest incidence of overall PC in the wealthy nations is that the Western diet is characterized by animal fat, high consumption of energy and meats, and low intake of fiber [67]. The incidence rate of PC is relatively lower in people living in India, China, Asia, and the East Mediterranean region [22]. Although the incidence rate of PC is lower in less developed countries, the mortality rate is higher when compared to wealthy nations. Meanwhile, the mortality rate of PC remains significantly high in predominantly black men worldwide than any other ethnic groups [66]. Several therapeutic options have been recommended to deal with the treatment of PC among AA men [68,69].

### 4.5. Socioeconomic Status in Prostate Cancer Disparity

It has been hypothesized that differences in PC disparity are associated with one’s socioeconomic status. The American Cancer Society predicts that one in every nine men living in the United States will be diagnosed with PC during their lifetime. This number increased with age, which suggests that about six in every 10 men in the United States aged 65 years or older will be diagnosed with PC over their lifetime [70]. AA men are disproportionally affected when compared to other racial groups [71,72]. One of the potential explanatory factors associated with PC disparity between Black/AA and Caucasian men is the inequality in socioeconomic status (SES) [73]. The SES could be measured by assessing the differences in the level of education and income, access to healthcare and fresh foods, and residential segregation [74]. Studies from the surveillance and epidemiology end results (SEER) database of men in California highlighted that PC patients with a low socioeconomic status usually have poor health insurance coverage, which may reduce their accessibility to cancer screening services and lead to late diagnosis, treatment, and worse mortality outcome [75,76]. Poverty was also associated with the lack of adequate health coverage, follow-up, and diagnosis in the advanced stage [76]. Weinrich and his collaborators reported that AA men have a low PC screening due to several barriers such as absence of health insurance, no sense of urgency, poor information on medical professional, cost of care, and lack of routine screening from a primary healthcare physician [77]. The majority of the black population compared to other ethnics/racial groups in the US stays in neighborhoods with poor quality and access to clean water and fresh food, transport, recreational facilities, and high insecurity [78,79]. This inaccessible to fresh food leads to their lower consumption of a plant-based diet. Meanwhile, fruits and vegetable possess phytochemicals and antioxidants that may prevent PC development [73,80]. On the other hand, lack of recreational parks and violence reduced the level of physical activity, outdoor activity, and increased the rate of obesity and stress [72].

### 4.6. Oxidative Stress and Age Factors in Prostate Cancer Disparity

The risk of developing PC increases gradually with age. Older men have the highest incidence of PC and African American (AA) men are disproportionately affected than any other ethnic groups [81,82]. The probability of developing PC has increased from 0.005% in males younger than 39 years to 2.2% in males between the ages 40 to 50 and approximately 13.7% in males between 60 to 79 years of age [83,84]. The PC is diagnosed at a younger age among AA men and they are approximately four times more likely to die from this disease compared to white men [82,85]. Based on this PC disparity, life expectancy of AA men is shorter compared to non-Hispanic white men in the United States [86]. The literature reveals a strong connection between oxidative stress, age, and PC. PC is commonly associated with a shift in the antioxidant-pro-oxidant balance towards increased oxidative stress. According to the free radical theory of aging, oxidative stress is blamed for aging because of the negative impacts excessive free radicals and/or the reactive oxygen species (ROS) have on the cells [87]. This phenomenon can be explained by the fact that older cells seem to be more susceptible to intracellular conditions that produce excess ROS, which induce tumorigenesis [88]. In addition, there is a decrease of antioxidant defense in the old population of men with PC [89]. Scientific data revealed that a shift in the pro-oxidant-antioxidant status in prostatic tissue of cells, animals, and humans played an important role in the initiation of prostate carcinogenesis [81]. 

## 5. Chemo-Preventive and Therapeutic Effects of Curcumin on Prostate Cancer

Curcumin (diferuloylmethane) is a polyphenolic compound derived from the rhizome of the plant *Curcumin longa* commonly known as turmeric. This compound has been widely used for centuries as a spice in Asia and the Middle East cuisine and as ayurvedic medicine due to its anti-oxidative, anti-inflammatory, and antiseptic properties [90]. Mounting evidence supports the unique chemo-preventive and therapeutic potential of curcumin. Many reports have shown that curcumin could inhibit initiation, progression, and metastasis of a variety of tumors including PC [91,92]. In PC, curcumin treatment has been shown to inhibit proliferation of both androgen-sensitive and androgen-independent PC cells [93,94]. Curcumin was shown to inhibit growth and proliferation of LNCaP and PC-3 cells *in vitro* [94,95] and in the xenograft model [94]. This inhibition of cell proliferation has been found to be associated with increased apoptosis [95,96,97]^.^ Curcumin treatment was also found to prevent metastasis of PC cells in many studies [98,99,100,101,102,103]. In addition, curcumin was demonstrated to inhibit angiogenesis in nude mice when they were injected with LNCaP and received a diet supplemented with curcumin [96]. Several scientific studies demonstrated that curcumin and its analogues exert anti-cancer effects on prostate cancer models including its effects on androgen receptor (AR) signaling and numerous downstream targets [104,105,106]. Another study indicated that curcumin analogues inhibited AR activity in prostate cancer cells [107]. Joseph et al. (2013) demonstrated that curcumin downregulated AR expression, limited AR binding to the androgen response element of the prostate specific antigen (PSA) gene, and reduced the expression of PSA in LNCaP cells [108]. Other studies demonstrated curcumin effectively delayed tumor growth and suppressed AR expression in an LNCaP xenograft model [109,110].

Some of the known anti-cancer effects of curcumin are mainly mediated through its negative regulation of a plethora of transcription factors, growth factors, inflammatory factors, protein kinases, and other oncogenic molecules. Curcumin was shown to down-regulate the expression and activity of androgen receptors [111,112] and co-factors [113] in LNCaP and PC-3 cells, which leads to growth inhibition. Curcumin was also shown to suppress the NF-kB activation and to downregulate the expression of both Bcl-2 and Bcl-xL in DU145 cells and LNCaP cells, which results in activation of both caspase-8 and caspase-3 as well as apoptosis [114]. Inhibition of cyclins [115,116] and upregulation of cyclin dependent kinase inhibitors [116] together with the increase in apoptosis were observed when PC cells were treated with curcumin. Curcumin was demonstrated to prevent metastasis in prostate cancer as well. When the metastatic C4-2B PC cells exhibit a tropism to bone and were injected in immunocompromised mice, curcumin treatment prevented the establishment of metastases in bone. This prevention appeared to be modulated by the induction of the expression of bone morphogenic protein-7 (BMP-7), which results in upregulation of the brown/beige adipogenic differentiation program implicated in the inhibition of bone metastasis [99]. Additional mechanisms by which curcumin prevents prostate metastasis involve suppression of cancer associated fibroblast-induced cell invasion [102], downregulation of the inflammatory cytokines CXCL1 and CXCL2 [117], and inhibition of the secretion of matrix metalloproteins [101,103]. In its control of angiogenesis, curcumin was found to significantly reduce micro-vessel density in a xenograft model. Curcumin has shown great promise in clinical trials as well [118]. Curcumin in combination with isoflavones was shown to reduce the levels of PSA in the serum of participants who had a serum PSA level above 10 ng/mL, which suggests a synergistic role of curcumin with isoflavones to reduce PSA production [119]. Many clinical trials are at different stages to investigate the therapeutic benefits of curcumin as a potential drug for many diseases. With its very limited side effects even at high doses and its pleitropic effects on many signaling pathways, curcumin represents an attractive drug that could be effective alone or in combination with other drugs in the management of multi-genic human diseases. 

## 6. Chemo-Preventive and Therapeutic Effects of Garlic on Prostate Cancer

Epidemiological studies suggest that high intake of garlic has protective effects against several human cancers including PC and the protection is even higher with increased intake [120]. The chemo-preventive effects of garlic are supported by several scientific studies. A report indicated that garlic possesses anti-cancer properties with the ability to prevent the growth of PC cells and enhance the immune system [121]. Garlic (*Allium sativum)* is a nutritious plant used for thousands of years as spices or seasoning, which is commercially available in the market in the form of garlic oil, garlic powder, or garlic extract [120,122]. It is also a medicinal agent that is composed mainly of water, carbohydrates, fiber, vitamin C, and minerals such as potassium, iron, magnesium, and phosphorus [122]. Flavonoids, saponins, proteins, and sapogenins are the most abundant phenolic compounds present in garlic [123,124]. Allicin, allixin, diallyl sulfide, diallyl disulfide, diallyl trisulfide, S-allylcysteinene allylmercaptan, allylmethyldisulfide, and ajoene are the dominant bioactive compounds present in garlic. These bioactive compounds act as chemo-preventive agents and their mechanisms of action include cell growth inhibition, cell cycle arrest, reactive oxygen species destruction, protein expression disruption, and induction of apoptosis [125,126,127]. These suggest that the consumption of garlic may be associated with a decreased risk of PC development. Epidemiological research from a population-based case-control study has revealed that men who consume garlic daily at the minimum dose of 10g/day have a lower risk of developing prostate cancer compared to men who consume less than 2.2 g/day [120]. Both *in vitro* and *in vivo* experiments have shown that garlic contains a specific anti-invasive compound capable of suppressing tumor growth in the prostate cancer cells and significantly prolonged life in the mouse test model [120,121]. Studies done by Kim and his colleagues showed that the diallyl trisulfide (DTAs) compound derived from garlic extraction induced apoptosis in the PC-3, LNCaP PC cell lines, and Transgenic Adenocarcinoma Mouse Prostate mice were exposed to 20 and 40 µmol/L for 8 and 16 hours [128]. In the same pattern, another research study found that DATs induced cell cycle arrest during the G2/M phase, decrease the expression of CDK1 and Cdc25C proteins, increase the cyclin B1, Bax, and Bak proteins in the PC-3 cells, and inhibit the intro cell migration [126,127,128]. A scientific report from our laboratory demonstrated that garlic extract inhibits the proliferation of human leukemia (HL-60) cells by producing malondialdehyde (a biomarker of cellular injury) and inducing apoptosis through the phosphatidylserine externalization and caspase-3 activation [129]. In summary, garlic could be viewed as a promising chemo-preventive agent against PC due to his ability to execute cell death through various signaling pathways.

## 7. Chemo-Preventive and Therapeutic Effects of *Vernonia amygdalina* on Prostate Cancer

*Vernonia amygdalina*, which is commonly called bitter leaf in English, is a medicinal plant that possesses many micro-nutrients important for the maintenance of human health and prevention of various diseases. It is widely consumed in Cameroon for nutritional purposes and has been used for many decades as ethno-medicine under the recommendation of herbalists for treating certain diseases and disorders including stomach discomfort, vomiting, malaria, diarrhea, and intestinal illnesses [130]. Furthermore, it has been used in Africa for preventing and treating diabetes, tuberculosis, measles, infertility in women, and hyperlipidemia [131,132,133]. Studies have revealed the presence of phytochemicals such as sesquiterpene lactones (vernolide, vernodalol), anthraquinones, flavonoids, lignans, phenolic acids, terpenes, and xanthones in *Vernonia amygdalina* and other *Vernonia* species [134,135]. These phytochemicals found in *Vernonia amygdalina* and other *Vernonia* species are good antioxidants. According to Khlebnikov et al. (2007), an antioxidant is any substance that can prevent, delay, or eliminate oxidative damage of a target molecule, and act as a regulator of the antioxidant defense [136]. Antioxidants from fruits and vegetables provide nutrients that support a healthy immune system and fight against cancer. At present, there are limited studies in the literature about the benefit of medicinal property of *Vernonia amygdalina* to PC patients. However, preliminary data and recently published studies from our laboratory have demonstrated that *Vernonia amygdalina* Delile acts as an antioxidant at a low dose and as pro-oxidant at a high dose when exposed to human prostate cancer (PC-3) cells [137]. In addition, we have demonstrated that the mechanisms involved in the chemoprevention of prostate cancer by *Vernonia amygdalina* Delile are mediated by inhibiting cell growth arrest, oxidative stress, DNA damage, and apoptosis, which is shown as phosphatidylserine externalization, activation of caspase-3, and cellular morphological changes [137]. Collectively, we have demonstrated the therapeutic effect of *Vernonia amygdalina* Delile on human prostate cancer (PC-3) cells. Our laboratory is currently testing the chemotherapeutic effects of *Vernonia amygdalina* Delile’s biologically active ingredient to elucidate its underlining mechanism of action *in vitro* and *in vivo*. Other *Vernonia* species such as *Vernonia divaricate* possess anticancer agents that significantly retard the growth of cancer cells including acute promyelocytic leukemia (HL-60) cells, human breast adenocarcinoma (MCF-7) cells, and PC-3 human prostate cancer cells [138]. Another study indicated that *Vernonia amygdalina* inhibited PC cells, but have no effect on normal human peripheral blood mononuclear cells [139]. Working with cell culture models and transgenic mouse models, our lab has also demonstrated that *Vernonia Amygdalina* exerts chemo-preventive action against breast cancer cells by targeting mechanisms that involve the arrest of cancer cell growth, the induction of DNA damage, and apoptosis accompanied by secondary necrotic cell death [129,130]. 

## 8. Summary of Chemo-Preventive and Therapeutic Effects of Curcumin, Garlic, and *Vernonia amygdalina* on Prostate Cancer

Curcumin, garlic, and *Vernonia amygdalina* are used worldwide in a variety of dishes. There are several epidemiological studies demonstrating their benefits and effects against cancers including PC. These medicinal plants exert their chemo-preventive and therapeutic effects on PC through several mechanisms of action (Table 1).

## 9. Molecular Mechanisms of Action of Medicinal Plants (Curcumin, Garlic, and *Vernonia amygdalina*)

Figure 3 shows the common molecular mechanisms by which curcumin, garlic, and *Vernonia amygdalina* exert their anti-cancer activity in cancer cells. As shown in Figure 3, curcumin, garlic, and *Vernonia amygdalina* inhibited cell growth or caused cell growth arrest through the induction of DNA damage, expression of stress genes (p53, p21, Bax, and CDK cyclin complex), impact of cell cycle arrest, and triggered apoptosis by modulating phosphatidylserine externalization, activation of caspase-3, and nucleosomal DNA fragmentation.

## 10. Conclusions

Prostate cancer (PC) remains a global public health issue in the United Sates, especially in African-American men who have a high mortality rate of PC compared to other racial groups. The treatment strategies for PC are limited, and chemoprevention is key to its future management. To overcome the burden of PC, scientists have made considerable progress by identifying plant-based foods that have chemo-preventive and therapeutic effects. In this research, we have reviewed extensively the biochemical properties and medicinal values of curcumin, garlic, and *Vernonia amygdalina* for their use in PC prevention and/or treatment. Consumption of these three medicinal plants contributes significantly to the prevention of PC. They are rich in vitamins, minerals, and phytochemicals such as flavonoids that are known to reduce the risk of PC. In addition, phytochemicals in these natural products are more likely to not only prevent PC development, but also reduce its incidence and mortality rates, improve the survival rate, and reduce racial disparity in PC.

## Figures and Tables

**Figure 1 nutrients-11-00336-f001:**
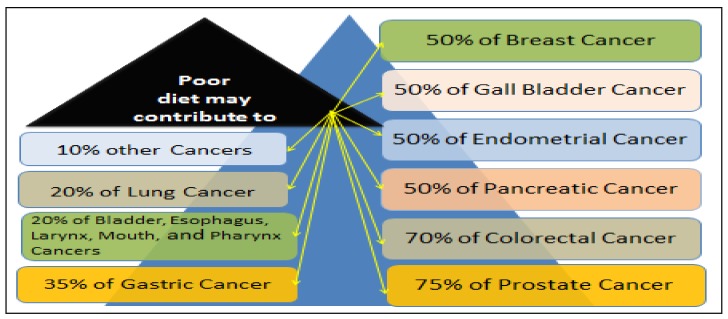
Estimation of cancer-related deaths associated with a poor diet. Eating a plant-based diet rich in vegetables and fruits has proven to significantly reduce cancer risk.

**Figure 2 nutrients-11-00336-f002:**
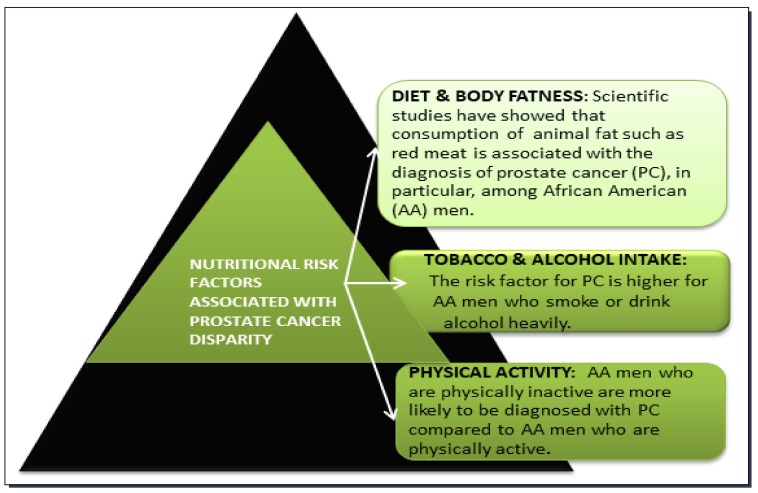
Nutritional risk factors associated with PC disparities. Poor diet, obesity, physical inactivity, excessive cigarette smoking, and high alcohol consumption seem to rise the overall risk of getting PC especially among AA men.

**Figure 3 nutrients-11-00336-f003:**
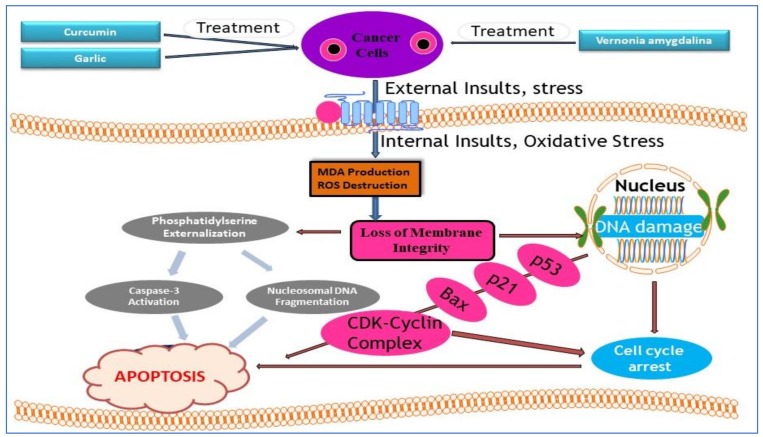
Schematic representation of common molecular mechanisms of curcumin, garlic, and *Vernonia amygdalina* as therapeutic agents in the management of malignancies.

**Table 1 nutrients-11-00336-t001:** Chemo-preventive and therapeutic effects of curcumin, garlic, and *Vernonia amygdalina* on prostate cancer.

Edible Medicinal Plants	Medicinal Uses	Pre-Clinical Studies	Clinical Studies	Summary & References
Curcumin	People use curcumin to prevent and/or treat cancer, Alzheimer, erectile dysfunction, baldness, hirsutism, and fertility	Inhibition of proinflammatory NF-B, reduction of prostate cancer growth, induction of apoptosis	Clinical trials needed	Curcumin has demonstrated the potential to slow growth and kill prostate tumor cells References [98,99,100,101,103,115].
Garlic (*Allium sativum)*	People use garlic to prevent and/or treat many conditions including prostate cancer, breast cancer, rectal cancer, stomach cancer, colon cancer, yeast infection, high blood pressure, hepatitis, and diabetes.	Cell growth arrest, cell cycle arrest, DNA damage, protein expression disruption, and induction of apoptosis	Few clinical trials have been performed, but they are still inconsistent. Recognition of garlic as one of the vegetables with potential anti-cancer properties	High consumption of garlic lowers the risk of prostate cancer. Sulfur present in garlic neutralizes cancer cells and shrinks the tumor References [120,121,126,128].
*Vernonia amygdalina*	People use *Vernonia amygdalina* to prevent and/or treat fever, hiccups, kidney problem, gastritis, enteritis, malaria, and rheumatism	Cell growth arrest, DNA damage, and induction of apoptosis as evidenced of phosphatidylserine externalization, activation of caspase-3, and cellular morphological changes.	Clinical trials needed	Limited scientific evidence. May act both as antioxidant and pro-oxidant depending on the dose. Further research needed. References [137,139].

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
