# Peer review of "Prostate Cancer Disparity, Chemoprevention, and Treatment by Specific Medicinal Plants"

_nutrients, 2019, doi:10.3390/nu11020336_

Round 1

Reviewer 1 Report

Here, authors present a review about PC disparity, and chemoprevention and treatment by medicinal plants. Although it is an interesting topic in PC, unfortunately authors did not provide detailed information about this topic.  In this review, the discussion is very crude.  Although tons of references were cited in this review, the authors did not take advantage of them.  Readers should be interested in the detailed data and information in these references, but author did not present the detailed information. 

Author Response

Dear Reviewer,

We would like to thank you for the thoughtful feedback and helpful comments.  To address the issues that were raised, we have streamlined and focused the manuscript considerably.  We have also gathered and compiled additional data so the changes that have been made are substantial and are intended to address all of the issues raised by the reviewer.  We trust that you will find that this is a significant improvement to the review. 

Specific comments

Reviewer Comments: Here, authors present a review about PC disparity and chemoprevention and treatment by medicinal plants. Although it is an interesting topic in PC, unfortunately, authors did not provide detailed information about this topic.  In this review, the discussion is very crude.  Although tons of references were cited in this review, the authors did not take advantage of them.  Readers should be interested in the detailed data and information in these references, but the author did not present the detailed information.  

Response to Reviewer Comment: Thank you for your comments on our review article.  We agree that the discussion is crude in some areas, so we have streamlined the manuscript to not only provide detailed, relevant and innovative information, but we also asked many of the contributors to provide inputs and editing assistance to improve the quality of the document, so we trust that these refinements will be self-evident in this revision.

Reviewer 2 Report

The authors could summarize the importance of these compounds in the form of a figure describing the effects on various signaling cascades and possible therapeutic targeting using combination therapy along with the well established drugs.

The report was interesting, however, a figure describing the pathways that can be targeted using these compounds would give great insight for pharmacological exploitation of the natural products and a table or a section including possible therapeutic approaches would be insightful along with any clinical trail data supporting the claims.

Author Response

Dear Reviewer,

We would like to thank you for the thoughtful feedback and helpful comments.  To address the issues that were raised, we have streamlined and focused the manuscript considerably.  We have also gathered and compiled additional data so the changes that have been made are substantial and are intended to address all of the issues raised by the reviewer.  We trust that you will find that this is a significant improvement to the review. 

Specific comments

Reviewer: The authors could summarize the importance of these compounds in the form of a figure describing the effects on various signaling cascades and possible therapeutic targeting using combination therapy along with the well-established drugs.

The report was interesting, however, a figure describing the pathways that can be targeted using these compounds would give great insight for pharmacological exploitation of the natural products and a table or a section including possible therapeutic approaches would be insightful along with any clinical trails data supporting the claims.

Response to Reviewer Comment: The reviewer’s input is extremely helpful, and we appreciate this feedback. As suggested, we carefully revised the manuscript and added entirely new content in the discussion section addressing common molecular/therapeutic mechanisms of curcumin, garlic, and Vernonia amygdalina in the management of cancer including PC. In addition, we added Figure 3 to summarize the possible signaling cascades and possible therapeutic targeting. 

Reviewer 3 Report

Comments and suggestions are listed below:

1. In the Introduction, etiology and disease progression to castration-resistant prostate cancer (CRPC), a lethal form of advanced disease, are missing. Also, please elaborate and address the current issues regarding the resistance to conventional therapy (e.g. ADT) and new strategy targeting CRPC. 

2. Section 4.1 (lines 132-153): Not all the studies indicated obesity as a risk factor for PCa. Please also list references demonstrating the controversial results. 

3. Section 5 (lines 209-247): Please elaborate AR targeting effects of curcumin extracts. AR overexpression, mutations, and AR splice variant expression are well-recognized phenotypes of advanced diseases with therapy resistance (e.g. Enzalutamide resistance). Thus, emphasis on anticaner effects of phytochemicals targeting AR function in PCa may benefit readers who read this review article. 

4. Line 227: PC-3 cell line is known for AR-negative. Please revise the manuscript accordingly.  

5. Section 7 (lines 280-313): Please elaborate oxidative stress in the context of CRPC progression. Oxidative stress condition is induced by ADT (please refer to FASEB J. 2017. Doi: doi.org/10.1096/fj.201601178R). Then, describe antioxidant therapies (including phytochemicals like polyphenols) preventing from disease progression to CRPC. 

Author Response

Dear Reviewer,

We would like to thank you for the thoughtful feedback and helpful comments.  To address the issues that were raised, we have streamlined and focused the manuscript considerably.  We have also gathered and compiled additional data so the changes that have been made are substantial and are intended to address all of the issues raised by the reviewer.  We trust that you will find that this is a significant improvement to the review. 

Specific comments

1.    Reviewer Comments: In the Introduction, etiology and disease progression to castration-resistant prostate cancer (CRPC), a lethal form of advanced disease, are missing. Also, please elaborate and address the current issues regarding the resistance to conventional therapy (e.g. ADT) and new strategy targeting CRPC. 

Response to Reviewer Comment: Thank you for this feedback.  We have addressed many of the points that you have raised about missing detailed information on etiology and disease progression to castration-resistant prostate cancer (CRPC) in our introduction. We have also made substantial changes to strengthen the paper. We have also added more details to our introduction to ensure that nothing that has been said is unsupported.  So your inputs have been extremely helpful and we appreciate this feedback. 

2.    Reviewer: Section 4.1 (lines 132-153): Not all the studies indicated obesity as a risk factor for PCa. Please also list references demonstrating the controversial results. 

Response to Reviewer Comment: The reviewer’s input is extremely helpful, and we appreciate this feedback. As suggested, we carefully revised the manuscript and added new content addressing the controversial studies on obesity risk factor associated with PCa.  

3.    Reviewer Comments: Section 5 (lines 209-247): Please elaborate AR targeting effects of curcumin extracts. AR overexpression, mutations, and AR splice variant expression are well-recognized phenotypes of advanced diseases with therapy resistance (e.g. Enzalutamide resistance). Thus, emphasis on the anticancer effects of phytochemicals targeting AR function in PCa may benefit readers who read this review article. 

Response to Reviewer Comment: The reviewer’s input is extremely helpful, and we appreciate this feedback. As suggested, we carefully revised the manuscript and added entirely new content in the discussion section addressing the elaborate AR targeting effects of curcumin and curcumin analogs.

Reviewer Comments: Line 227: PC-3 cell line is known for AR-negative. Please revise the manuscript accordingly.  

Response to Reviewer Comment: Thank you for this helpful suggestion.  We revised the manuscript as part of our overall effort to streamline this paper and make it more focused.

4.    Reviewer Comments: Section 7 (lines 280-313): Please elaborate oxidative stress in the context of CRPC progression. Oxidative stress condition is induced by ADT (please refer to FASEB J. 2017. Doi: doi.org/10.1096/fj.201601178R). Then, describe antioxidant therapies (including phytochemicals like polyphenols) preventing from disease progression to CRPC. 

Response to Reviewer Comment: We agree with this critique and added new entire content on oxidative stress in the context of CRPC progression as part of our overall effort to streamline this review and make it more focused.

Round 2

Reviewer 1 Report

Authors have addressed the issues. It can be published.